# Identification and Characterization of Endophytic Fungus DJE2023 Isolated from Banana (*Musa* sp. cv. Dajiao) with Potential for Biocontrol of Banana Fusarium Wilt

**DOI:** 10.3390/jof10120877

**Published:** 2024-12-17

**Authors:** Longqi Jin, Rong Huang, Jia Zhang, Zifeng Li, Ruicheng Li, Yunfeng Li, Guanghui Kong, Pinggen Xi, Zide Jiang, Minhui Li

**Affiliations:** Guangdong Province Key Laboratory of Microbial Signals and Disease Control, College of Plant Protection, South China Agricultural University, Guangzhou 510642, China; jlq@stu.scau.edu.cn (L.J.); 13797509760@163.com (R.H.); bzwtla@126.com (J.Z.); anime801lsa@gmail.com (Z.L.); 19849089840@163.com (R.L.); yunfengli@scau.edu.cn (Y.L.); gkong@scau.edu.cn (G.K.); xpg@scau.edu.cn (P.X.); zdjiang@scau.edu.cn (Z.J.)

**Keywords:** banana cv. Dajiao, endophytic fungi, pathogenicity, antagonistic activity, biological control, banana Fusarium wilt

## Abstract

This study characterized an endophytic fungus, DJE2023, isolated from healthy banana sucker of the cultivar (cv.) Dajiao. Its potential as a biocontrol agent against banana Fusarium wilt was assessed, aiming to provide a novel candidate strain for the biological control of the devastating disease. The fungus was isolated using standard plant tissue separation techniques and fungal culture methods, followed by identification through morphological comparisons, multi-gene phylogenetic analyses, and molecular detection targeting *Fusarium oxysporum* f. sp. *cubense* (*Foc*) race 1 and race 4. Furthermore, assessments of its characteristics and antagonistic effects were conducted through pathogenicity tests, biological trait investigations, and dual-culture experiments. The results confirmed isolate DJE2023 to be a member of the *Fusarium oxysporum* species complex but distinct from *Foc* race 1 or race 4, exhibiting no pathogenicity to banana plantlets of cv. Fenza No.1 or tomato seedlings cv. money maker. Only minute and brown necrotic spots were observed at the rhizomes of banana plantlets of ‘Dajiao’ and ‘Baxijiao’ upon inoculation, contrasting markedly with the extensive necrosis induced by *Foc* tropical race 4 strain XJZ2 at those of banana cv Baxijiao. Notably, co-inoculation with DJE2023 and XJZ2 revealed a significantly reduced disease severity compared to inoculation with XJZ2 alone. An in vitro plate confrontation assay showed no significant antagonistic effects against *Foc*, indicating a suppressive effect rather than direct antagonism of DJE2023. Research on the biological characteristics of DJE2023 indicated lactose as the optimal carbon source for its growth, while maltose favored sporulation. The optimal growth temperature for this strain is 28 °C, and its spores can germinate effectively within the range of 25–45 °C and pH 4–10, demonstrating a strong alkali tolerance. Collectively, our findings suggest that DJE2023 exhibits weak or non-pathogenic properties and lacks direct antagonism against *Foc*, yet imparts a degree of resistance against banana Fusarium wilt. The detailed information provides valuable insight into the potential role of DJE2023 in integrated banana disease control, presenting a promising candidate for biocontrol against banana Fusarium wilt.

## 1. Introduction

Bananas (*Musa* spp.) are an important fruit and staple crop that hold a crucial position in global cultivation and consumption. However, banana production is facing significant challenges, particularly from Fusarium wilt caused by the soilborne fungus *Fusarium oxysporum* f. sp. *cubense* (*Foc*) tropical race 4 [1]. The pathogen can survive for decades in the soil in the form of chlamydospores. When host banana plantlets are planted in the infested soil, *Foc* infects the roots and colonizes the rhizomes or pseudostems, leading to wilting and death of the plants [2]. This significantly impacts the yield of bananas, making disease control extremely challenging. In orchards severely affected by Fusarium wilt, traditional susceptible cultivars of Cavendish banana can no longer be cultivated, highlighting the urgent need to explore disease control methods [3].

At present, the control of banana Fusarium wilt involves several methods, including the selection and breeding of disease-resistant varieties, the application of biological control methods, the implementation of agricultural control measures, and the formulation of comprehensive prevention and control strategies [4]. Although breeding disease-resistant cultivars is the most economically effective approach, it faces significant challenges due to the difficulty in the genetic transformation of bananas, their triploid nature that prevents conventional hybrid breeding, and the public’s reluctance to accept genetically modified plants [5,6]. Therefore, the current focus should be on biological control, particularly by identifying endophytes from bananas that exhibit antagonistic effects against *Foc*. This would provide new materials for the control of banana Fusarium wilt.

Endophytic fungi are a special class of microorganisms that form either symbiotic or non-symbiotic relationships with plants and typically do not cause noticeable disease symptoms [7]. These fungi play a significant role in promoting the growth of their host plants by enhancing nutrient absorption and preventing the invasion of plant diseases and pests through mechanisms such as nutrient competition and antagonism [8,9]. Previous research has demonstrated that endophytic fungi isolated from the rhizosphere and plant tissues of banana plants have positive effects on improving plant resistance against *Foc*, root-knot nematodes, and frost damage caused by low temperatures [10,11,12,13,14,15]. This indicates the potential for their application in biological control.

This study characterized an endophytic fungus, DJE2023, isolated from healthy banana sucker of the cultivar ‘Dajiao’. Its potential as a biocontrol agent against banana Fusarium wilt was assessed, aiming to provide a novel candidate strain for the biological control of the devastating disease.

## 2. Materials and Methods

### 2.1. Isolation and Preservation of Endophytic Fungus

Healthy suckers of banana cv. Dajiao, at the development of the floral differentiation stage, were collected from an orchard located in Machong Town, Dongguan, Guangdong Province, China. The rhizomes of the healthy suckers were cut and washed under tap water and then cut into small pieces of 3–5 mm^3^ for the isolation of endophytic fungi using conventional microbe isolation methods [16]. The tissues were placed on potato dextrose agar (PDA) plates and incubated at 28 °C for 3–5 days. Newly grown fungal colonies were then transferred to fresh PDA plates. The resulting fungal isolate was named DJE2023, preserved in 20% glycerol and stored at −80 °C.

### 2.2. Plant Material and Fungal Inoculation

In this study, three banana cultivars, Fenza No.1, Dajiao, and Baxijiao, as well as tomato cv. money maker, were used for fungal inoculation. Tissue culture-derived banana plantlets at the 5–7 leaf stage and tomato seedlings germinated for 15 days were inoculated using the root dipping inoculation assay as described previously [17]. Fungal isolate DJE2023 was inoculated in yeast extract peptone dextrose (YPD) media and incubated at 28 °C for 2 days. The conidia were collected, washed with ultrapure water, and resuspended in ultrapure water to a final concentration of 10^5^ spores/mL. Thirty plantlets of each cultivar were selected, and their roots were soaked in the conidia suspension for 30 min. Water treatment was used as a negative control, and *Foc* tropical race 4 strain XJZ2 was set as a positive control. The inoculated plantlets were planted in home gardening nutrient soil after treatment and placed in a growth chamber as described previously [17]. Disease symptoms were assessed 42 days after inoculation by the browning degree of the rhizomes dissected longitudinally. The disease severity was recorded according to the disease grading criteria, and the disease index was calculated using the following formula. The disease grading criteria used were as follows: grade 0 indicated no vascular discoloration was observed; grade 1 indicated a little vascular discoloration expressed as brown dots; grade 2 indicated that vascular discoloration accounted for up to 50%; grade 3 indicated that vascular discoloration was over 50% in the corm of the banana plantlet [18].
$$\mathrm{Disease}\;\mathrm{Index}=\frac{\sum \left(\mathrm{Number}\;\mathrm{of}\;\mathrm{plantlets}\times \mathrm{Grade}\;\mathrm{value}\right)}{\mathrm{Total}\;\mathrm{number}\;\mathrm{of}\;\mathrm{plantlets}\times \mathrm{Maximum}\;\mathrm{grade}\;\mathrm{value}}\times 100$$

### 2.3. Identification of Fungal Isolate DJE2023

To observe and compare the colony morphology and microscopic morphology, the endophytic fungal isolate DJE2023 was cultured on a PDA plate for 7 days. Microscopic morphology, including microconidia, macroconidia, and chlamydospores, was examined using a microscope. Photomicrographs were taken, and the sizes of these structures were measured.

DNA was extracted from the fungal isolate DJE2023 using the OMEGA DNA extraction kit. PCR amplification was performed using housekeeping genes, including *Tef1*, *Rpb2*, *CmdA*, and ITS conserved sequence primers (Table A1) [19]. The PCR products were purified to obtain the target DNA fragments, which were then cloned into the pMD 18 T-vector for sequencing, and the obtained sequences were used for phylogenetic analysis. Phylogenetic trees were constructed using the maximum likelihood method in RAxML software (version 8.2.12) according to the method described by Stamatakis A. [20].

To further confirm the identity of the isolate, specific detection primers for *Foc* race 1 (W1805F/W1805R), *Foc* tropical race 4 (W2987F/W2987R), and universal primers (W106F/W106R) were used in the study [19]. PCR amplification with these specific primers was conducted, and the PCR products were subjected to agarose gel electrophoresis. The race of the isolate was confirmed by the presence of the target band similarly to the positive control of race-identified isolates.

### 2.4. Antagonistic Cultivation

On PDA plates, DJE2023 was co-cultured with the DJ-2023-3 (*Foc* race 1) and XJZ2 (*Foc* TR4) strains to assess whether DJE2023 exhibits antagonistic activity against the pathogens. A 0.5 cm diameter plug of DJE2023 and equally sized plugs of DJ-2023-3 and XJZ2 were placed at one-third positions on the PDA plates, and the plates were incubated at 28 °C in the dark. Photographs were taken at 4 days post-co-cultivation [21].

### 2.5. Biological Characteristics of Endophytic Fungi

Effects of different carbon sources on the growth and sporulation of DJE2023: MM medium was supplemented with 20% glucose, fructose, maltose, sucrose, mannitol, or lactose as the sole carbon source (15 mL/plate). A 6 mm diameter DJE2023 mycelial plug was placed in the center of each plate and incubated in darkness at 28 °C for 5 days. Colony morphology, colony diameter, and spore production were observed and measured. Each treatment was repeated three times.

Effects of different temperatures on DJE2023 growth: A 6 mm diameter DJE2023 mycelial plug was inoculated onto PDA plates (15 mL/plate) and incubated in darkness at 16 °C, 20 °C, 24 °C, 28 °C, 32 °C, and 36 °C for 5 days. Colony diameter was measured for each condition. Each treatment was repeated three times.

Effects of different temperatures on DJE2023 spore germination: A DJE2023 spore suspension was prepared at a concentration of 10^3^ spores/mL and dispensed into centrifuge tubes. The tubes were incubated in water baths at 25 °C, 30 °C, 35 °C, 40 °C, 45 °C, and 50 °C for 10 min. An amount of 100 μL of each suspension was spread onto PDA plates (15 mL/plate) and incubated in darkness at 28 °C for 2 days. The number of individual colonies was recorded. Each treatment was repeated three times.

Effects of different pH levels on DJE2023 growth: PDA medium was adjusted to pH levels of 4, 5, 6, 8, and 10 before sterilization. A 6 mm-diameter DJE2023 mycelial plug was inoculated onto each plate, which was then incubated in darkness at 28 °C for 5 days. Colony morphology and diameter were observed and measured. Each treatment was repeated three times.

### 2.6. Statistical Analysis

All statistical analyses were performed using IBM SPSS Statistics (Version 27) [22]. All data were disposed as mean ± standard deviation (S.D.) and analyzed by Duncan’s Multiple Range test at *p* = 0.01 and *p* = 0.05 significance levels. Differences were considered to be significant at *p* < 0.05 and very significant at *p* < 0.01. Figures were drawn using GraphPad Prism v 6.01 (GraphPad Software, Inc., San Diego, CA, USA).

## 3. Results

### 3.1. Morphological Characteristics of Endophytic Fungus DJE2023

A strain of endophytic fungus, designated DJE2023, was isolated from healthy banana suckers. Its colonies appear light pink and fluffy with a pale red secretion (Figure 1a). Under the microscope, colorless oval-shaped microconidia, either single-celled or double-celled, were observed. The single-celled conidia were measured to be approximately 3.20 to 4.53 μm × 1.07 to 1.60 μm (n = 50). Additionally, colorless macroconidia, sickle-shaped with three to eight septa, ranged in size from 12.8 to 33.9 μm (n = 50) (Figure 1b). The chlamydospores were spherical in shape, with diameters approximately ranging from 9.20 to 9.32 μm (n = 50) (Figure 1c).

### 3.2. Molecular Identification and Phylogenetic Analysis

Since the endophytic fungal strain DJE2023 was isolated from banana plants, it was necessary to determine whether it is a strain of *Foc*, the causal agent of banana Fusarium wilt disease. Race 1-specific primers W1805F/W1805R, race 4-specific primers W2987F/W2987R, and universal primers W106F/W106R were used to amplify the genomic DNA of three strains, including *Foc* race 1 and race 4, as well as the endophytic fungus DJE2023 [19]. The results are shown in Figure 1. The positive controls for *Foc* race 1 and race 4 produced bands at 354 and 729 bp, and 593 and 729 bp, respectively. In contrast, the endophytic fungus DJE2023 only exhibited a single band at 729 bp, with no amplification in the negative control using water (Figure 1e). These results suggested that DJE2023 was not a typical *Foc* strain.

To further identify the taxonomic position of DJE2023, a multi-gene phylogenetic tree was constructed using sequences from ITS (538 bp), *CmdA* (617 bp), *Rbp2* (953 bp), and *Tef1* (701 bp). A comparative analysis of DJE2023 with 19 other Fusarium strains generated a phylogenetic tree (Figure 1d). The strains and gene information used to construct the phylogenetic tree are listed in Table A2. The results revealed that DJE2023 clusters closely with the *F. oxysporum* species complex, including *F. oxysporum* f. sp. *cubense* race 1 and tropical race 4 isolates, sharing a high bootstrap value of 99. It is indicated that DJE2023 belonged to the *F. oxysporum* species complex, showing the closest genetic affinity to the banana Fusarium wilt pathogen, but it does not match the specific races of *Foc* tested in this study.

### 3.3. Pathogenicity Tests

The root dipping method was employed to assess the pathogenicity of the fungal strain DJE2023 on plantlets of three banana cultivars: Baxijiao, Dajiao, and Fenza No.1. After 42 days of incubation, the results showed that ‘Fenza No.1’ remained largely free from yellowing, with healthy roots and no browning at the base of the rhizome. In contrast, ‘Baxijiao’ and ‘Dajiao’ exhibited mild browning in the rhizomes but the symptoms were significantly less severe compared to those induced by the positive control, the *Foc* tropical race 4 strain XJZ2. The XJZ2 resulted in severe leaf yellowing, wilting, and complete plant mortality after 28 days (Figure 2a).

The comparison of the symptoms indicated that DJE2023 did not fully exhibit the classic symptoms associated with *Foc*, suggesting weak or absent pathogenicity. The average disease indices for ‘Baxijiao’, ‘Dajiao’, and ‘Fenza No.1’ inoculated with DJE2023 were 5.83, 9.17, and 0.00, respectively (Figure 2b). This indicated that DJE2023 exhibited low pathogenicity to ‘Baxijiao’ and ‘Dajiao’ and was non-pathogenic to ‘Fenza No.1’. In comparison, the average disease index for ‘Baxijiao’ inoculated with XJZ2 was 79.17 (Figure 2c), highlighting the significantly stronger pathogenicity of XJZ2.

Additionally, approximately 5 × 10^5^ spores/mL of DJE2023 were inoculated to thirty tomato seedlings. After 22 days, no leaf yellowing was observed, and the stem sections showed intact vascular bundles without browning, further confirming the non-pathogenic nature of DJE2023 in tomatoes (Appendix A).

In another trial, ‘Baxijiao’ plantlets were inoculated individually and jointly with spore suspensions of XJZ2 and DJE2023 at a concentration of approximately 5 × 10^6^ spores/mL. After 45 days, banana leaves inoculated solely with XJZ2 showed partial yellowing, longitudinal rhizome sections, and pronounced browning at the base of the rhizome with a spreading trend. In contrast, DJE2023 single inoculation resulted in minimal yellowing of banana leaves, with robust and healthy root systems, unaffected rhizome sections, and no browning at the rhizome base. Leaves inoculated with either XJZ2 or DJE2023 exhibited mild yellowing and limited browning, while the negative control group showed no signs of disease (Figure 3a).

The disease survey revealed that the proportion of severe disease and disease index in the mixed inoculation group were significantly lower than those observed in the XJZ2-inoculated group, indicating that DJE2023 could attenuate the pathogenicity of the host bananas (Figure 3b,c).

To investigate the incidence of banana Fusarium wilt in banana plantations from which the DJE2023 was isolated, we conducted follow-up field surveys approximately every two months from June 2023 to March 2024. Throughout the growing season, no symptoms of Fusarium wilt were observed in the ‘Dajiao’ plants (Table A3).

### 3.4. Relationship Between Endophytic Fungi and Pathogenic Fungi

To explore the interaction between endophytic and pathogenic fungi, DJE2023 was co-cultured with banana wilt pathogen physiological races 1 and 4 on PDA plates. As depicted in Appendix A, DJE2023 showed slightly faster growth than banana wilt pathogen physiological races 1 and 4. The relationship between endophytic fungi and pathogenic fungi exhibited only a competitive interaction, without a significant antagonistic effect.

### 3.5. Effects of Carbon Sources, Temperature, and pH on the Growth and Sporulation of Endophytic Fungus DJE2023

Investigating the impact of various carbon sources on DJE2023 revealed that lactose promoted the fastest growth (8.3 cm colony diameter), while maltose resulted in the highest sporulation (1.9 × 10^6^ spores/mL). Sorbitol led to the slowest growth (5.9 cm) and lowest sporulation (5 × 10^5^ spores/mL). Fructose and sucrose also caused slower growth and lower sporulation compared to lactose and maltose (Figure 4a–c). Optimal growth occurred within the temperature range of 24–28 °C, with 28 °C being the most favorable, while growth was minimal at 16 °C (3 cm colony diameter) (Figure 5a,b). Between 25 and 45 °C, spores exhibited the highest germination rates, whereas at 50 °C, the germination rates were significantly lower, with only a few spores germinating (Appendix A). DJE2023 thrived at pH 10, demonstrating notable alkali resistance, but could not grow at pH 4 (Figure 6a,b).

## 4. Discussion

Endophytic fungi are known to form either symbiotic or non-symbiotic relationships with plants and typically do not cause noticeable disease symptoms [23]. However, under certain conditions, such as when the plant is wounded or under stress, some endophytes can transition into pathogens [8,24]. In our study, when DJE2023 was inoculated onto the banana plantlets of cv. Dajiao and Baxijiao, only minute and brown necrotic spots were observed at the rhizomes, which contrasts markedly with the extensive necrosis induced by the positive control, *Foc* tropical race 4 strain XJZ2. Additionally, DJE2023 did not cause any wilt symptoms when inoculated in plantlets of banana cv. Fenza No.1 and tomato seedlings. A follow-up disease survey in the field showed that banana cv. Dajiao plants, the original host of DJE2023, exhibited no symptoms of Fusarium wilt throughout the growing season. These results suggest that DJE2023 is capable of causing minor symptoms under artificial injury inoculation conditions, and it does not exhibit significant pathogenicity. Therefore, we classified DJE2023 as a fungal endophyte rather than a pathogen.

Plants show phenotypic plasticity in response to changing or extreme abiotic environments; however, over millions of years, they also have co-evolved to respond to the presence of soil microbes [25]. As a fungal endophyte, DJE2023 is likely to promote the development of beneficial microbial communities around the roots of bananas by establishing a symbiotic relationship with the plant. Studies have shown that endophytic fungi can form symbioses with host plants, effectively preventing pathogen invasion and spread by occupying root space and acquiring nutrients and other survival resources [26]. DJE2023’s rapid growth characteristics and tolerance to alkaline conditions may give it a competitive advantage in the microbial community of the banana rhizosphere. Especially in soils with significant pH fluctuations, these traits of DJE2023 facilitate its better adaptation and stable colonization in the environment, thereby reducing the infection opportunities of pathogens such as *Foc* TR4 through resource competition and niche exclusion. Therefore, the application of DJE2023 not only leverages its direct biocontrol potential but also indirectly suppresses pathogens by modifying the rhizosphere microbial environment and enhancing the diversity of beneficial microorganisms, offering a new strategy for the integrated management of banana wilt disease.

Biological control has been widely used and a large number of biocontrol agents have been selected as primary strategies for suppressing soil-borne fungal disease [27]. The microbiome from banana roots or rhizosphere soil has been reported to be a major source of biocontrol agents for controlling banana Fusarium wilt [21,28,29]. In this study, using these strategies, we screened and identified a fungal endophyte, DJE2023, as a member of *F. oxysporum* species complex (FOSC), which is a highly diverse group of *Fusarium* species encompassing strains with various ecological types and host specificities [30]. Non-pathogenic *F. oxysporum* strains have been reported previously in tomato and banana [12,31]. For example, an endophytic *F. oxysporum* strain, Fo47, first isolated from healthy tomato in 1987, was verified as a non-pathogenic fungus [31]. Subsequent studies found that Fo47 had suppressive effects on Fusarium wilt, primarily attributed to microbial antagonism and a certain degree of systemic induced resistance, which may be related to the induced expression of PR-1 and accumulation of chitinase [32].

In this study, DJE2023 showed significant resistance to Fusarium wilt in pot trail, and no direct antagonistic activity was observed in vitro, implying its potential application as a biocontrol agent through indirect mechanisms. The weak pathogenicity of DJE2023 towards bananas and its evident disease resistance effect suggests that it might inhibit pathogen infection by inducing systemic resistance (ISR) in the plant. Many studies have shown that endophytic fungi can enhance the host plant’s defense capabilities by triggering ISR, making the plant more resistant to pathogen invasion [33]. For instance, *Trichoderma citrinoviride* HT-1 primed *Rheum palmatum* roots against root rot fungus *Fusarium oxysporum* by increasing the expression of JA/SA responsive genes (AOC, LOX, PR5, PR4, and PR1) and stimulating higher activities of the antioxidant enzymes (SOD, POD, CAT, and PAL) [33,34]. Therefore, DJE2023 may enhance the plant’s immune response and induce systemic resistance in bananas, effectively suppressing the disease manifestation of *Foc* even in the absence of direct antagonistic action. However, the mechanism of resistance activation of DJE2023 still needs to be investigated.

Based on our previous research, six virulent microRNA-like small RNAs (milRNAs) derived from *Foc* have been identified to play significant roles in the initial infection process [35,36,37]. These virulent milRNAs provide precise targets for the efficient control of banana Fusarium wilt. In addition, recent studies have shown that endophytic fungi can transfer RNA interference (RNAi) signals to host plants, regulating the gene expression of the host during mutualistic symbiosis, which may lead to beneficial phenotypic changes [38]. In this study, DJE2023, a fungal endophyte isolated from healthy suckers of the banana cv. Dajiao demonstrated a natural advantage in colonizing banana plants. This suggests that DJE2023 could be a more effective biocontrol agent compared to other endophytes from alternative host plants. To develop DJE2023 into a practical biological control agent, future research needs to evaluate its efficacy and feasibility under field conditions. Specifically, studies focusing on soil treatment, the formulation of the fungal inoculum, and application methods will be crucial for optimizing its control performance. By serving as an RNAi vector, DJE2023 can deliver specific milRNAs to the pathogen or target genes in the plant, thereby interfering with the gene expression of the pathogen and reducing its virulence. Future work will focus on developing silencing vectors targeting these selected virulent milRNAs, which can be more effectively delivered to plants through the carriage of DJE2023, achieving targeted silencing of pathogenic genes and effectively reducing the pathogenicity of the pathogens. This strategy not only enhances the disease resistance of the host plant but also reduces reliance on chemical fungicides, thereby minimizing environmental pollution and ecological impacts.

## 5. Conclusions

In conclusion, this study identified an endophytic *F. oxysporum* strain, DJE2023, from the healthy suckers of the banana cv. Dajiao. Pot trials demonstrated that this strain can effectively control banana Fusarium wilt, although not through traditional antagonistic mechanisms. We also analyzed its biological characteristics, which will help to utilize the endophyte in the field, providing a new avenue for the management of banana Fusarium wilt. Further research is needed to explore the underlying mechanisms of its suppressive effects and to develop practical applications for this promising biocontrol agent.

## Figures and Tables

**Figure 1 jof-10-00877-f001:**
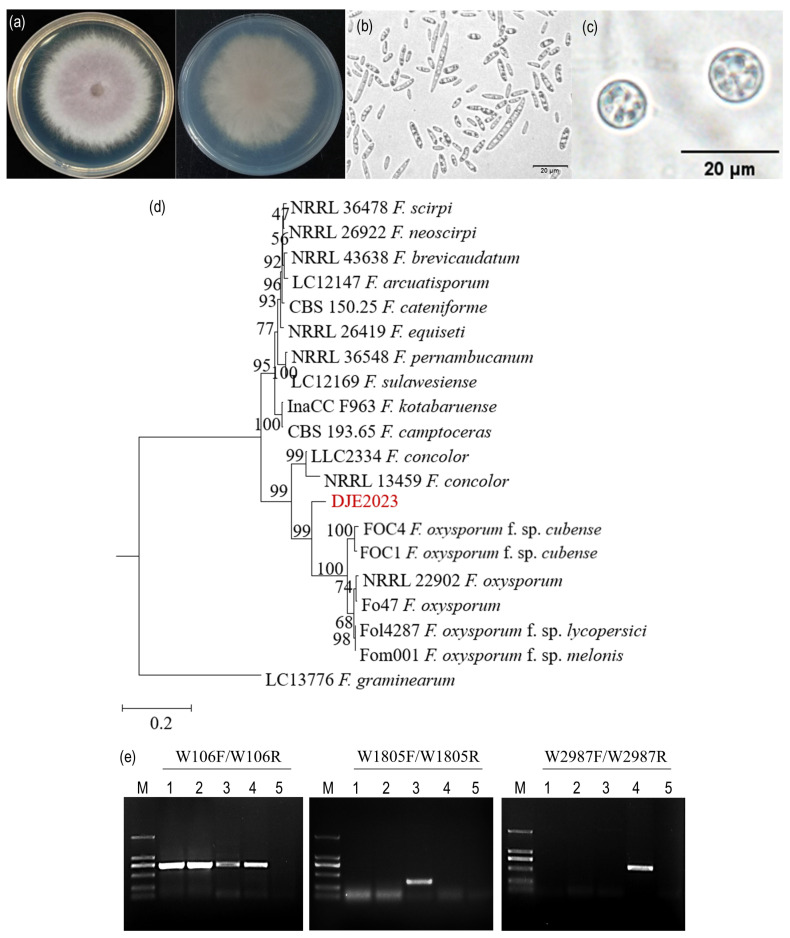
Identification of endophytic fungus DJE2023 from banana cv. Dajiao. (**a**) The colony morphology of DJE2023. (**b**) Conidia. (**c**) Chlamydospores. (**d**) Multigene system phylogenetic tree. (**e**) Molecular identification using specific primers for *Fusarium oxysporum* f. sp. *cubense* (*Foc*) races 1 and 4. Among them, W106F/W106R are the universal primers for *F. oxysporum*, W1805F/W1805R are the specific primers for race 1of *Foc*, and W2987F/W2987R are the specific primers for race 4 of *Foc*. M: DL2000 Marker; 1,2: DJE-2023; 3: FOC Race 1; 4: FOC Race 4; 5: Water control.

**Figure 2 jof-10-00877-f002:**
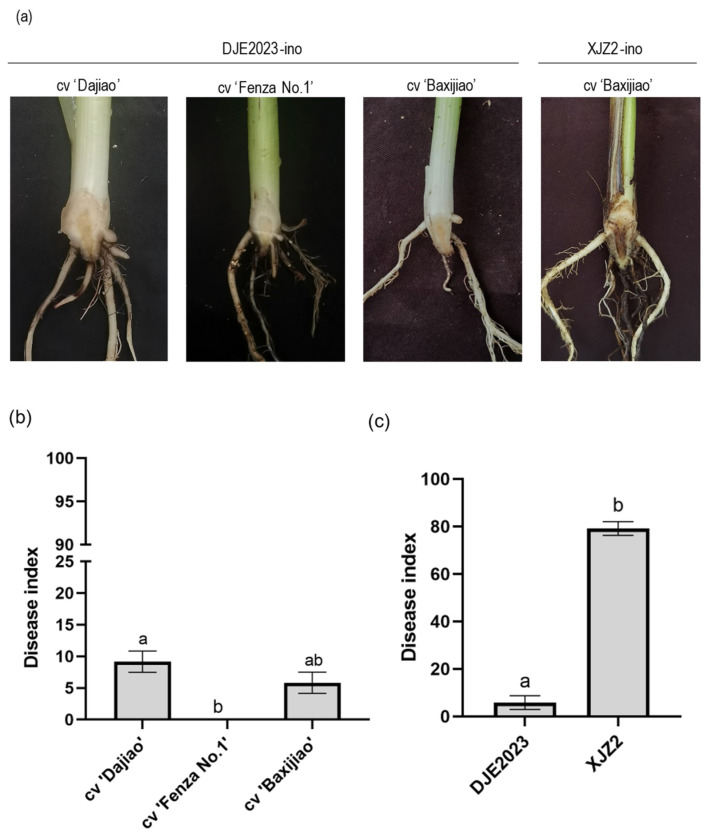
The pathogenicity test results for the banana endophytic strain DJE2023 and the *Fusarium oxysporum* f. sp. *cubense* (*Foc*) race 4 strain XJZ2. (**a**) The typical symptoms exhibited in the rhizomes of various banana cultivars. (**b**) The disease incidence statistics resulting from inoculating strain DJE2023 onto various banana cultivars. (**c**) The disease incidence statistics resulting from inoculating DJE2023 and XJZ2 onto banana cv. Baxijiao. The inoculation data of 30 plantlets were randomly divided into three groups for statistical analysis and the mean ± S.D. (n = 3). The significant difference (Duncan test, *p* < 0.05) between two groups is represented by the superscript letters above the error bars.

**Figure 3 jof-10-00877-f003:**
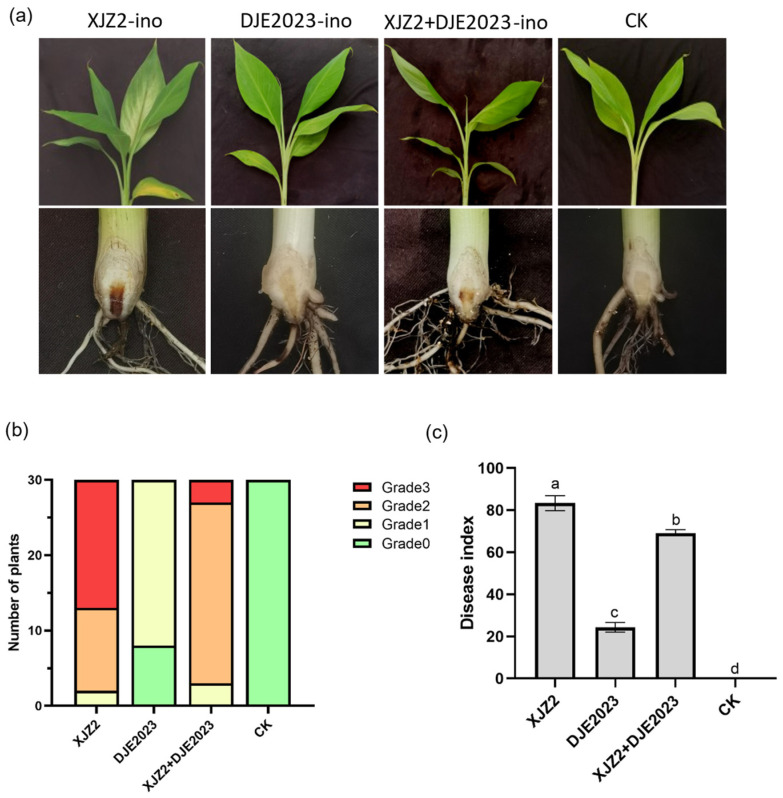
The biocontrol effect of DJE2023 on the banana plantlets cv. Baxijiao. (**a**) XJZ2, DJE2023, XJZ2, and DJE2023 co-infect the roots of banana plantlets; CK is water control. The disease conditions of leaves and roots are observed 45 days later. (**b**) Disease severity levels of banana roots; each treatment was replicated with 30 banana plantlets. (**c**) Disease index analyzed by diseased plantlet number of different disease grades. The inoculation data of 30 plantlets were randomly divided into three groups for statistical analysis and the mean ± S.D. (n = 3). The significant difference (Duncan test, *p* < 0.05) between two groups is represented by the superscript letters above the error bars.

**Figure 4 jof-10-00877-f004:**
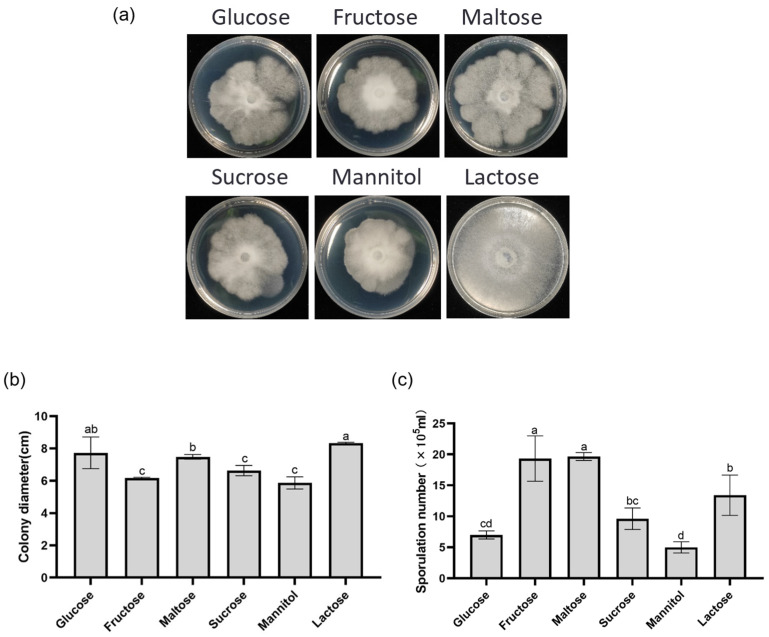
Effects of carbon source on colony morphology, growth diameter, and sporulation of DJE2023. (**a**) The effect of carbon sources on the colony morphology of DJE2023. (**b**) The effect of carbon sources on the growth diameter of DJE2023. (**c**) The effect of carbon sources on the spore production of DJE2023. All data are presented as the mean ± standard deviation (n = 3). The significant difference (Duncan test, *p* < 0.05) between two groups is represented by the superscript letters above the error bars.

**Figure 5 jof-10-00877-f005:**
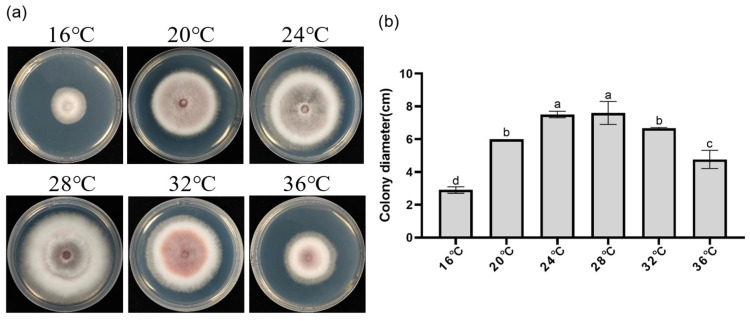
Effect of temperature on the morphology and growth diameter of DJE2023 colonies. (**a**) Colony morphology of DJE2023 under different temperatures. (**b**) Effect of temperature on the growth diameter of DJE2023. All data are presented as the mean ± standard deviation (n = 3). The significant difference (Duncan test, *p* < 0.05) between two groups is represented by the superscript letters above the error bars.

**Figure 6 jof-10-00877-f006:**
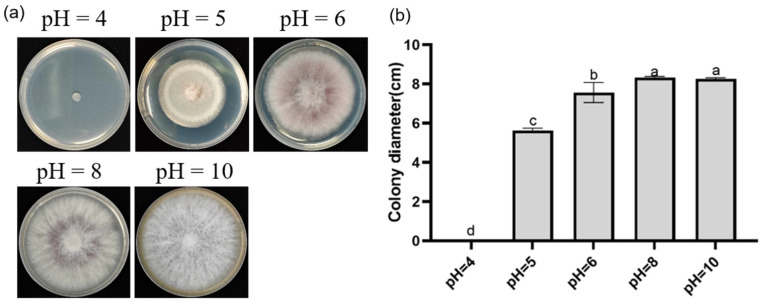
Effect of pH on colony morphology and growth diameter of DJE2023. (**a**) Colony morphology of DJE2023 under different pH levels. (**b**) Effect of pH on the growth diameter of DJE2023. All data are presented as the mean ± standard deviation (n = 3). The significant difference (Duncan test, *p* < 0.05) between two groups is represented by the superscript letters above the error bars.

## Data Availability

The original contributions presented in this study are included in the article; further inquiries can be directed to the corresponding author.

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
