# Peer review of "Identification and Characterization of Endophytic Fungus DJE2023 Isolated from Banana (*Musa* sp. cv. Dajiao) with Potential for Biocontrol of Banana Fusarium Wilt"

_jof, 2024, doi:10.3390/jof10120877_

Round 1

Reviewer 1 Report

The comments are in the attached file.

The comments are in the attached file.

Reviewer 2 Report

The results shown in this work have a failure in the conclusion, since   the Pathogenicity Tests are quite worrying because there is a lesion in the two banana cultivars, however small, already indicates some susceptibility of the host to the isolated strain of fusarium. I consider that this makes it unviable to use this strain to control this banana disease. Perhaps it could be a candidate for another phytopathogenic fungus. 

Moreover, although a competitive action does not disqualify a microorganism for use as a biocontrol agent, it is somewhat alarming that there is no inhibition of the pathogen and that, in turn, the Fusarium strain DJE2023 generates certain lesions in the pathogenicity tests. 

In my opinion, it is not acceptable to use this biocontrol strain in this pathosystem.

The classification of DJE2023 as endophytic fungus needs much more evaluation. I consider that the results shown in this paper do not indicate this.

The figures are ok. 

Reviewer 3 Report

In this article, a series of experiments were conducted to study the biocontrol properties of the endophytic fungus Fusarium oxysporum DJE2023 against pathogenic strains of the same species that cause banana Fusarium wilt. This research is important for agriculture. The advantage of this work is the use of an integrated approach, in which experiments conducted both in vitro and in vivo. A particularly interesting experiment is the one with joint treatment of plants with endophytic and pathogenic strains shown in Figure 3. This experiment demonstrates that the endophyte has a competitive advantage over the pathogen, although it does not completely lack pathogenic properties. One disadvantage of this study is the lack of genetic research on these strains, which would help link the antagonistic properties of the endophytic strain with its genotype characteristics. This would aid in understanding the mechanism of competition between the strains. Nevertheless, the results obtained convincingly demonstrate the anti-pathogenic properties of the endophytic fungus with respect to pathogenic strains.

Lines 306-308. How can small RNAs be targeted to other small RNAs, reducing their expression? Are there any studies that show this mechanism?

Figure 1d, Table A1. Please write the GenBank access numbers of the DNA sequences in the text.

Reviewer 4 Report

The manuscript "Identification and Characterization of Endophytic Fungus DJE2023 Isolated from Banana (Musa sp. cv 'Dajiao') with Potential for Biocontrol of Banana Fusarium Wilt" by Longqi Jin, Rong Huang, Jia Zhang, Zifeng Li, Ruicheng Li, Yunfeng Li, Guanghui Kong, Pinggen Xi, Zide Jiang and Minhui Li describes a new method for effective control of banana Fusarium wilt. 

The relevance of the study is well described in the introduction.

The results are statistically significant. They are well described and illustrated.

The Discussion section seems too short to me, which is supported by a relatively small list of references.

The materials and methods are described in detail, however, I did not find a detailed method for culturing plants in reference 17.

Round 2

Reviewer 2 Report

After the exhaustive review carried out by the authors, I consider that the manuscript has improved and that the weak points of the work in the results and conclusion have been satisfactorily justified.

no comments

Author Response

Thank you for your positive comments. 

We sincerely thank you for your thorough review of our manuscript and your valuable comments. Your suggestions have played a crucial role in improving the quality of our paper.